# Combined Transcriptomic and Metabolomic Analysis Reveals the Role of Phenylpropanoid Biosynthesis Pathway in the Salt Tolerance Process of *Sophora alopecuroides*

**DOI:** 10.3390/ijms22052399

**Published:** 2021-02-27

**Authors:** Youcheng Zhu, Qingyu Wang, Ying Wang, Yang Xu, Jingwen Li, Shihui Zhao, Doudou Wang, Zhipeng Ma, Fan Yan, Yajing Liu

**Affiliations:** College of Plant Science, Jilin University, Xi’an Road, Changchun 130062, China; yczhu19@mails.jlu.edu.cn (Y.Z.); qywang@jlu.edu.cn (Q.W.); wangying2009@jlu.edu.cn (Y.W.); yangxu16@mails.jlu.edu.cn (Y.X.); jingwen@jlu.edu.cn (J.L.); shzhao18@mails.jlu.edu.cn (S.Z.); wangdd18@mails.jlu.edu.cn (D.W.); mazp20@mails.jlu.edu.cn (Z.M.)

**Keywords:** *Sophora alopecuroides*, salt stress, transcriptome, metabolome, phenylpropanoid biosynthesis, lignin, flavonoids

## Abstract

Salt stress is the main abiotic stress that limits crop yield and agricultural development. Therefore, it is imperative to study the effects of salt stress on plants and the mechanisms through which plants respond to salt stress. In this study, we used transcriptomics and metabolomics to explore the effects of salt stress on *Sophora alopecuroides*. We found that salt stress incurred significant gene expression and metabolite changes at 0, 4, 24, 48, and 72 h. The integrated transcriptomic and metabolomic analysis revealed that the differentially expressed genes (DEGs) and differential metabolites (DMs) obtained in the phenylpropanoid biosynthesis pathway were significantly correlated under salt stress. Of these, 28 DEGs and seven DMs were involved in lignin synthesis and 23 DEGs and seven DMs were involved in flavonoid synthesis. Under salt stress, the expression of genes and metabolites related to lignin and flavonoid synthesis changed significantly. Lignin and flavonoids may participate in the removal of reactive oxygen species (ROS) in the root tissue of *S. alopecuroides* and reduced the damage caused under salt stress. Our research provides new ideas and genetic resources to study the mechanism of plant responses to salt stress and further improve the salt tolerance of plants.

## 1. Introduction

Soil salinization is a global problem. More than 80 million ha of land globally are affected by salinization, which accounts for approximately 6% of the global land area [1,2]. Of the 1.5 billion ha of dryland agriculture, approximately 32 million ha are affected by soil salinization, and this figure is increasing [3]. As one of the main abiotic stresses that restrict plant growth and development, salt stress severely restricts the development of agriculture and affects grain yield and quality [3]. Therefore, studying the effects of salt stress on plants and exploring the mechanism of plant salt tolerance are of great significance to discover salt tolerance-related genes, develop and utilize salt-tolerant plant resources, and improve plant salt tolerance [4].

The main effects of salt stress on plants are osmotic stress and ion toxicity. Osmotic stress is mainly due to the low water potential around plant roots, which leads to cell dehydration that affects cell division and expansion, and it further affects plant growth and development [1]. Ion toxicity is caused by excessive salt entering the transpiration stream of plants, causing damage to the cells in the leaves, and affecting the photosynthetic ability of plants [5,6]. Moreover, the Na^+^ that plants are forced to absorb under high salt stress conditions accumulates gradually and inhibits the absorption of K^+^ that is an element necessary for plant growth and the absence of which reduces plant productivity [7]. The excessive accumulation of reactive oxygen species (ROS) can damage essential substances such as DNA, proteins, and lipids, which affect the function of plant cells, inhibit the normal growth of plants, and, in severe cases, cause plant death [8]. Reducing or eliminating the influence of ROS on plant cells may be a breakthrough point to further improve plant salt tolerance.

As fixed organisms, plants cannot escape the external environment or its negative influences [5]. Plants can only adapt to the environment by changing their forms [5]. Salt stress originating in soil forces plants to evolve an adaptive system, which includes various physiological and biochemical mechanisms, which are collectively known as the plant salt stress response [8]. Thus far, the plant salt stress response components that have been discovered include ion homeostasis and division, ion transport and absorption, osmotic protectant biosynthesis, antioxidant enzyme and antioxidant compound synthesis, the synthesis of active molecules such as polyamines, hormone regulation, and changes in membrane structure [9,10]. The accumulation of a variety of soluble osmotic adjustment substances under salt stress conditions can reduce water loss under short-term osmotic stress, enhance cell turgor, stabilize cell structure under long-term osmotic stress conditions, and facilitate the continuous growth of cells [11]. The ROS produced by plants under salt stress can activate enzymatic and non-enzymatic systems to alleviate oxidative stress [12]. The enzyme system consists of antioxidant enzymes, such as peroxidase (POD), catalase (CAT), superoxide dismutase (SOD), glutathione reductase (GR), and glutathione S-transferase (GST), which remove the ROS system [13]. Non-enzymatic systems are metabolites produced in response to salt stress and include ascorbic acid, alkaloids, flavonoids, carotenoids, glutathione, and phenolic compounds [8,13,14]. The role played by the enzymatic and the non-enzymatic systems is mainly to eliminate ROS to reduce its damage to plant cells and improve the antioxidant capacity of plants [12,14]. The products of the phenylpropane metabolic pathway play a certain role in the process of eliminating ROS. The three hydroxycinnamyl alcohol precursors in the phenylpropanoid biosynthesis (PHB) pathway, namely p-coumaroyl, coniferyl alcohol, and sinapyl alcohol, derive p-hydroxybenzyl (H-lignin), guaiacyl (G-lignin), and syringyl lignin (S-lignin), respectively [15]. Lignin, a product of lignin synthesis pathway (LSP), is a dietary fiber that is insoluble in water. The lignin content in plants is second only to cellulose, and it affects plants’ tolerance to abiotic stress [16]. Flavonoids are derivatives of chalcones, such as flavanones (e.g., naringenin), flavonols (e.g., quercetin), isoflavones (e.g., formononetin), and others [17]. A variety of different chemical classes of flavonoids directly or indirectly participate in signal transduction pathways that regulate oxidative stress and cell survival to varying degrees [18]. Flavonoids have a wide range of physiological functions in the tolerance response of plants to abiotic stress, and they play an important role in the elimination of hydrogen peroxide and ROS [19,20,21,22]. Chalcone synthase (CHS), chalcone isomerase (CHI), and cytochrome P450 monooxygenase (CPM) are key enzymes in the flavonoid synthesis pathway (FSP) and play key roles in plant salt stress [23,24,25]. Arabidopsis *AtCHS* and soybean *GmCHS* play positive regulatory roles in plant responses to salt stress, while CPM and CHI play negative regulatory roles [26]. Studies have found that genes in the PHB pathway are involved in the process of plant response to salt stress, but there are few reports on the changes of each gene in response to plant response to salt stress and the relationship between each gene [23,24,25,26].

With the development of science and technology, omic technologies have gradually matured. Employing omic technologies is an efficient technical means to study the effects of salt stress on plants and the mechanism of plant salt tolerance. The current research efforts on plant responses to salt stress are mainly concentrated in the model plant genus Arabidopsis, followed by rice, corn, wheat, and other crops [27,28,29,30,31]. However, the mining and utilization of salt-tolerant plant resources need to be explored further. There are fewer genetic resources for the application of plant stress tolerance genes and the effective improvement of plant salt tolerance. The mining and utilization of stress tolerance genes for legumes are equally reduced. *Sophora alopecuroides* (Fabaceae) is a leguminous perennial herb in the genus Robinia; it is an excellent sand-fixing pioneer plant distributed mainly in the desert and semi-desert areas of Northwest China [32]. The morphological characteristics of *S. alopecuroides* are pinnate compound leaves, oval-shaped leaves, terminal racemes, white or light yellow corolla, and beaded pods (Appendix A). It is a plant exhibiting excellent resistance to stress, thereby having a certain development and utilization value for stress resistance studies [32]. However, there are few reports on the research regarding the stress resistance genes of *S. alopecuroides*, and the whole genome of *S. alopecuroides* has not been completed. In the present study, we employed joint transcriptomic and metabolomic analyses to identify salt tolerance genes in *S. alopecuroides* and explore the role of PHB pathway genes in response to salt stress. This research has a certain guiding significance for studying the influence of salt stress on plants and the analysis of plant salt tolerance mechanisms, and it provides more choices and possibilities for improving crop salt tolerance through genetic engineering technology.

## 2. Results

### 2.1. Physiological Changes of S. alopecuroides under Salt Stress

To explore the physiological changes of *S. alopecuroides* under salt stress, we treated 4-week-old seedlings with 1.2% NaCl. The results showed that the leaves of *S. alopecuroides* showed mild wilting after 4 h under the salt treatment and gradually recovered within 24 h; however, the leaves withered again at 72 h. The physiological indicators of *S. alopecuroides* treated with salt were measured at 0, 4, 24, 48, and 72 h (Figure 1). Fresh leaf samples were used to determine the relative water content (RWC), malondialdehyde (MDA), chlorophyll, soluble sugar, proline, hydrogen peroxide (H_2_O_2_), peroxidase (POD), catalase (CAT), and superoxide dismutase (SOD) values, while root tissue was used to determine the Na^+^ and K^+^ contents. The RWC of the leaves of the salt-treated plants decreased gradually over time. After 24 h, the RWC of the salt-treated plants was significantly different from that of the control plants (*p* < 0.01). The MDA content gradually increased with the accumulation of salt stress and increased significantly at 72 h. The chlorophyll content showed a trend consistent with phenotypic changes under salt stress. The soluble sugar content was initially upregulated and then downregulated under salt stress. The proline content increased with the accumulation of salt stress. The H_2_O_2_ content was higher 4 h and 72 h into the salt treatment. POD and SOD exhibited the highest activity 24 h into the salt treatment, and CAT exhibited the highest activity at 72 h. The flame spectrophotometry results showed that both the Na^+^ and K^+^ contents continued to increase under salt stress, but the K^+^/Na^+^ ratio gradually decreased. Additionally, we found significant physiological differences in *S. alopecuroides* 4 h after exposure to salt stress. While the physiological changes at 24 h and 48 h were small, these were obvious at 72 h. This result was consistent with the observed phenotypic changes. *S. alopecuroides* may suffer from osmotic stress at the initial stage (<4 h) under salt stress, and then its phenotype gradually recovers. This may be due to the gradual adaptation of *S. alopecuroides* to salt and osmotic stress.

### 2.2. Transcriptomic Analysis of the Response of S. alopecuroides to Salt Stress

Detailed transcriptional sequencing information is shown in Appendix A. The principal component analysis (PCA) that was performed on the gene expression levels of the samples (Figure 2A) showed that salt stress induced changes in the genes of *S. alopecuroides.*

The gene annotation success rate statistics are presented in Table 1. Comparing annotations with the Nr database can allow us to obtain information on related species with similar gene sequences as *S. alopecuroides*. Our results showed that *S. alopecuroides* had higher similarity with *Lupinus angustifolius*, followed by *Cajanus cajan* (Figure 2B, Evalue_distribution, Similarity_distribution in Appendix A).

Genes successfully annotated by GO (Gene Ontology) were classified into the next level according to the three major GO categories (biological process (BP), cellular component (CC), and molecular function (MF)) (Appendix A). The results showed that in BP, the most enriched genes were in the metabolic process (GO:0008152), cellular process (GO:0009987), biological regulation (GO:0065007), and biological process regulation (GO:0050789). In CC, the most enriched genes were in cellular anatomical entities (GO:0110165), intracellular (GO:0005622), and the protein-containing complex (GO:0032991). In MF, genes were mainly enriched in binding (GO:0005488) and catalytic activity (GO:0003824). The most annotated genes in the KOG (euKaryotic Ortholog Groups) database were in translation, ribosomal structure, and biogenesis (4914), followed by post-translational modification, protein turnover, chaperones (4050), and general function prediction only (3089) (Figure 2C).

Gene expression level analysis was performed on the five sets of transcriptomic data (three biological replicates per group), and significance analysis was performed for each comparison combination (DESeq2 padj < 0.05 |log2FoldChange| > 1) to obtain differentially expressed genes (DEGs) (Appendix A). The differential gene volcano map produced based on our results is shown in Appendix A. To screen for genes with higher expression levels and obvious differences, we determined the parameter again (corrected_*p*_value < 0.05 |log2FoldChange| > 2) and counted the number of DEGs obtained in different salt treatment time points (Figure 3A).

The statistical results of metabolic pathways annotated in the KEGG (Kyoto Encyclopedia of Genes and Genomes) database for DEGs obtained during different salt stress treatment periods (Appendix A, Venn diagram as shown in Figure 3B) show that there were 49 metabolic pathways jointly annotated with DEGs at various periods. Our results showed that the primary metabolic pathways with more annotated DEGs were mainly the sugar, amino acid, and nucleic acid metabolism pathways. Sugar metabolism includes glycolysis/gluconeogenesis (ko00010), the citrate cycle [Tricarboxylic acid (TCA) cycle] (ko00020), the pentose phosphate pathway (ko00030), pentose and glucuronate interconversions (ko00040), galactose metabolism (ko00052), starch and sucrose metabolism (ko00500), and amino sugar and nucleotide sugar metabolism (ko00520). Amino acid metabolism included alanine, aspartate, and glutamate metabolism (ko00250), glycine, serine, and threonine metabolism (ko00260), cysteine and methionine metabolism (ko00270), valine, leucine, and isoleucine degradation (ko00280), valine, leucine, and isoleucine biosynthesis (ko00290), lysine biosynthesis (ko00300) and degradation (ko00310), arginine and proline metabolism (ko00330), histidine metabolism (ko00340), tyrosine metabolism (ko00350), phenylalanine metabolism (ko00360), tryptophan metabolism (ko00380), and phenylalanine, tyrosine, and tryptophan biosynthesis (ko00340). Nucleic acid metabolism pathways included purine (ko00230) and pyrimidine metabolism (ko00240). Energy metabolism-related factors included oxidative phosphorylation (ko00190) and carbon fixation in photosynthetic organisms (ko00710). The secondary metabolic pathways with many annotated genes mainly included diterpenoid biosynthesis (ko00904), brassinosteroid biosynthesis (ko00905), carotenoid biosynthesis (ko00906), zeatin biosynthesis (ko00908), phenylpropanoid biosynthesis (ko00940), and flavonoid biosynthesis (ko00941). Other pathways included nitrogen metabolism (ko00910), plant hormone signal transduction (ko04075), and plant–pathogen interaction (ko04626).

Among the aforementioned pathways, those that were significantly enriched for DEGs included ko00500, ko00904, ko00906, ko00908, ko00910, ko00940, ko00941, ko04075, and ko04626 (Figure 3C). The results showed that these pathways were closely correlated to the response of *S. alopecuroides* to a salt stress environment (q_value < 0.05).

In the present study, we used mainstream hierarchical clustering to perform cluster analysis on gene FPKM (expected number of Fragments Per Kilobase of transcript sequence per Millions base pairs sequenced) values and selected eight clustering results that showed obvious trends (upregulated and downregulated) with the accumulation of salt stress for metabolic pathway analysis. The results showed that the differentially expressed genes were significantly enriched (*p* < 0.05) in the metabolic pathways ko00500, ko00940, ko04075, and ko04626 (Figure 3D). These pathways provide us with new ideas for further mining the salt tolerance genes of *S. alopecuroides.*

### 2.3. Metabonomic Analysis on the Response of S. alopecuroides to Salt Stress

To further explore the changes in the metabolic level of *S. alopecuroides* under salt stress conditions, we selected *S. alopecuroides* roots treated with salt at 0 h (CK), 24 h (T24), 48 h (T48), and 72 h (T72) for metabolite analysis. The metabolites of 24 samples (six biological replicates in each group) were analyzed by PCA at the expression level (Figure 4A). The results showed that the roots of *S. alopecuroides* exhibited significant metabolic changes under salt stress.

We obtained a total of 1098 metabolites, of which 666 were detected in the positive ion mode and 432 were detected in the negative ion mode. The obtained metabolites were annotated using the KEGG pathway. The results showed that the metabolites obtained under positive and negative ion mode conditions were mainly annotated in the global and overview maps (pos: 68, neg: 66), the biosynthesis of other secondary metabolites (pos: 36, neg: 32), amino acid metabolism (pos: 28, neg: 28), carbohydrate metabolism (pos: 6, neg: 19), nucleotide metabolism (pos: 13, neg: 10), and the metabolism of cofactors and vitamins (pos: 13, neg: 6) (Appendix A).

The screening of differential metabolites (DMs) mainly refers to variable importance in the projection (VIP), fold change (FC), and the *p*-value. The VIP value represented the contribution of metabolites to the group, FC was the ratio of the average value of each metabolite in all biological replicates in the comparison group, and the *p*-value was calculated using a *t*-test [33], which represents the level of difference. We set thresholds (VIP > 1.0, FC > 1.5, or FC < 0.667, and a *p*-value < 0.05) to screen for DMs (Appendix A) [34,35,36]. T24_vs_CK had 188 DMs (158 upregulated and 30 downregulated); T48_vs_CK had 237 DMs (165 upregulated and 72 downregulated); T72_vs_CK had 309 DMs (213 upregulated and 96 downregulated 96). These results indicate that with the accumulation of salt stress, the DMs obtained by *S. alopecuroides* gradually increased, and the differential metabolism was more upregulated than downregulated (Figure 4B).

Hierarchical clustering analysis (HCA) [37] was performed on all the DMs of the obtained comparison pairs, and the relative quantitative values of the DMs were normalized and clustered (Figure 4C). The results showed that the metabolites changed significantly in the salt-treated groups.

### 2.4. Integrated Transcriptomic and Metabolomic Analysis of the Response of S. alopecuroides to Salt Stress

To reveal the expression of the transcriptional regulation of the state of the salt stress *S. alopecuroides* genes, the transcriptome and metabolome were compared for correlation analysis. In this study, we selected the T24_vs_CK, T48_vs_CK, and T72_vs_CK groups to obtain significant differentially expressed genes (DEGs) and DMs (*p* < 0.05) for correlation analysis, using Pearson’s correlation coefficient to measure the degree of correlation between DEGs and DMs (when the correlation coefficient k < 0, the correlation was negative; when k > 0, the correlation was positive) (Figure 5A). DEGs and DMs in the significantly enriched pathways were selected for correlation analysis. In T24_vs_CK, we analyzed the correlation between 74 DEGs and 20 DMs; in T48_vs_CK, we analyzed the correlation between 54 DEGs and 24 DMs; in T72_vs_CK, we analyzed the correlation between 100 DEGs and 28 DMs.

All DEGs and DMs were mapped to the KEGG pathway database, their common pathway information was obtained, and we determined the main biochemical pathways and signal transduction pathways that DMs and DEGs participated in together (Figure 5B). The results showed that in T24_vs_CK, the commonly annotated and significantly enriched pathways were ko04075, ko00940, and ko00500; in T48_vs_CK, they were ko00940, ko00500, and ko00941; in T72_vs_CK, they were ko00940, ko00941, and ko00945 (stilbenoid, diarylheptanoid, and gingerol biosynthesis). This indicated that secondary metabolic pathways gradually dominated the response of *S. alopecuroides* to salt stress, as the latter accumulated.

### 2.5. Combined Transcriptomic and Metabolomic Analysis on the Response of S. alopecuroides in the LSP under Salt Stress

The combined transcriptomic and metabolomic analysis revealed that the DEG and DM pathways that were significantly enriched were the LSP and the FSP. We further analyzed the expression of DEGs and DMs in these two pathways under salt stress conditions. In the LSP, 28 DEGs in eight families were found to be significantly related to seven different metabolites, including phenylalanine ammonia-lyase (PAL), trans-cinnamate 4-monooxygenase (C4H), caffeic acid 3-O-methyltransferase (COMT), 4-coumarate–CoA ligase (4CL), cinnamoyl-CoA reductase (CCR), cinnamyl alcohol dehydrogenase (CADH), peroxidase (POD), and coniferyl-alcohol glucosyltransferase (UGT) (Figure 6A,B). Seven different metabolites were detected, including L-phenylalanine, trans-cinnamic acid, ferulic acid, cinnamaldehyde, caffeic aldehyde, sinapyl alcohol, and coniferin (Figure 6C). DMs exhibited a mainly gradual upregulation trend with the accumulation of salt stress. Among them, the upstream L-phenylalanine, trans-cinnamic acid, and ferulic acid were mainly upregulated and then downregulated with the accumulation of salt stress, with the highest performance at 48 h. DEGs related to these metabolites showed expression changes that were initially upregulated, then downregulated, and finally upregulated, with the highest expression level at 24 h. This result led us to hypothesize that there was a time difference between the expression of genes and the formation of metabolites in *S. alopecuroides*. We found that downstream DMs, such as cinnamaldehyde, caffeic aldehyde, sinapyl alcohol, and coniferin continued to increase under salt stress conditions (Figure 6C).

In this pathway, most of the DEGs identified belonged in the peroxidase family. Among them, *SaPOD6, SaPOD9,* and *SaPOD10* were significantly upregulated at various salt stress periods (Figure 6B). This result indicates that the LSP may be one of the main *S. alopecuroides* salt stress response pathways; additionally, the lignin produced during this process may be the main *S. alopecuroides* substance produced in response to salt stress.

### 2.6. Combined Transcriptomic and Metabolomic Analysis on the Response of S. alopecuroides in the FSP under Salt Stress

Flavonoids are among the main secondary metabolites of plants and play an important role in the response of plants to abiotic stress. The combined transcriptomic and metabolomic analysis revealed that 24 DEGs, including shikimate O-hydroxycinnamoyltransferase (HCT), chalcone synthase (CHS), chalcone isomerase (CHIS), coumaroylquinate (coumaroylshikimate) 3′-monooxygenase (CYM), flavonoid 3′,5′-hydroxylase (FHD), flavonol synthase (FLS), anthocyanidin reductase (ANR), and leucoanthocyanidin dioxygenase (ANS), were obtained in the FSP (Figure 7A,B). The seven different metabolites were naringenin chalcone, naringenin, vitexin, 5-O-caffeoylshikimic acid, isoliquiritigenin, pelargonidin chloride, and quercetin (Figure 7C). Seven DEGs belonged to the HCT; these were initially upregulated, then downregulated, and finally upregulated, while they exhibited the highest expression levels at 4 h and 72 h. Eight DEGs were obtained from the CHS family; their expression level trend was consistent with that of the HCT family, but the expression level of each gene was the highest at 24 and 72 h. This result indicates that there were differences in the timing of the response of different genes to salt stress, which may be caused by differences in the expression order of different genes in response to salt stress.

In the present study, we found that the salt treatment induced water loss and wilting in *S. alopecuroides* leaves at 4 h, phenotypic recovery at 24 h, and withering again at 72 h. The DEGs obtained in the LSP and FSP were initially upregulated, then downregulated, and finally upregulated again. This result is consistent with changes in plant phenotypes. We suspect that plants in the early stage of salt stress activate the upregulation of genes in the LSP and then produce metabolites that respond to salt stress. In the mid stages of salt stress, the state of the plant tends to be stable, and the expression of salt-responsive genes is downregulated. However, *S. alopecuroides* was still highly resistant to salt stress, owing to the accumulation of metabolites. After 72 h, salt accumulation in *S. alopecuroides* was higher, leading to the upregulation of salt tolerance genes in response to salt stress.

To further verify the dynamic expression of DEGs in response to salt stress and the reliability of the transcriptomic results, we selected *SaPAL2, SaC4H3, SaCOMT1, Sa4CL3, SaCCR, SaCADH, SaPOD5, SaUGT, SaHCT2, SaCYM, SaFHD, SaANS, SaCHS5, SaCHIS2, SaANR2,* and *SaFLS* for qRT-PCR verification (Figure 8). The qRT-PCR results were consistent with the transcriptomic analysis results.

## 3. Discussion

*Sophora alopecuroides* is a species that is highly resistant to stress; therefore, examining and utilizing its genetic material is significant in researching plant resistance. Phenotypic changes in plants under salt stress can be observed, but their physiological changes need to be further experimentally explored to study the effects of salt stress on plants. In this study, RWC was used as an index to measure the water condition and osmotic balance of plants under stress, as it can reflect the degree of damage incurred by salt stress to plants [38]. The RWC of *S. alopecuroides* gradually decreased under salt stress, but the rate of change also decreased gradually, thereby indicating that *S. alopecuroides* may reduce the damage caused by salt stress through self-regulation. MDA is a product of lipid peroxidation and serves as a marker of plasma membrane damage [39]. The results of this study showed that the MDA content of the leaves of *S. alopecuroides* under salt stress was higher than that of the control leaves, and the increase was less from 4 h to 48 h, thereby indicating that *S. alopecuroides* prevented the destruction of its plasma membrane by salt stress at this stage. Under salt stress, Na+ accumulates in plant leaves, disrupts the chloroplast structure, affects photosynthesis, and slows plant growth [38]. We measured the chlorophyll content of salt-treated *S. alopecuroides* and found that the recovery upregulation occurred at 24 h and 48 h, thereby indicating that *S. alopecuroides* reduced the influence of salt stress on photosynthesis. Sugars are photosynthates and respiration substrates [39,40], and they provide a carbon skeleton and energy for plant growth. An increase in the soluble sugar content improves cell osmotic regulation and protoplasm protection [34,38]. Soluble sugar may also be used as a signaling molecule to participate in plant salt tolerance [41]. The soluble sugar content of *S. alopecuroides* significantly increased after 4 h of salt stress, thereby indicating that sugar may play a role in the process of salt stress. ROS, as a secondary stress substance in salt stress, causes secondary damage to plants, but at the same time, it activates the plant antioxidant response and produces antioxidant enzymes (POD, SOD, and CAT) that help remove excessive ROS from plant cells in order to maintain the redox balance of plant tissue cells [42]. The H_2_O_2_ content in *S. alopecuroides* treated with salt was initially upregulated, then downregulated, and finally upregulated; additionally, the antioxidant enzyme content was significantly higher than that of the control group, thereby indicating that the ROS scavenging mechanism may be activated. Na^+^ and K^+^ change significantly when plants are exposed to salt stress. Salt stress significantly increases the Na^+^ content and reduces the K^+^ content of alfalfa [43]. High Na^+^ concentrations lead to a decrease in K^+^ in plant cells, thereby causing an imbalance in cell osmosis and hindering plant growth [44,45]. Our results showed that the Na^+^ content of the salt-treated *S. alopecuroides* increased significantly. This result is consistent with the changes observed in *alfalfa, corn* and other plants under salt stress. However, we found that the K^+^ content in the root tissue of *S. alopecuroides* was upregulated within a short salt stress treatment period. This may be due to the response of *S. alopecuroides* aimed at maintaining osmotic balance under salt stress. This provides a reference for the further exploration and analysis of the salt tolerance mechanisms of *S. alopecuroides*.

Plants respond to salt stress through complex regulatory mechanisms [43]. The transcriptome and metabolome have been widely used to study the effects of salt stress on plants and the regulatory mechanisms of plants in response to salt stress, including *Jerusalem artichoke* [46], *sweet potato* [47], *cotton* [48], *cucumber, Beta vulgaris* [49], *poplar species* [50], *spinach* [51], *Zygophyllum* [52], *Medicago truncatula* [53], *tomato* [54], *barrey* [55], *wild soybean* [56], *peanut, Casuarina glauca* [57], and *lotus* [58,59]. The DEGs and DMs obtained under salt stress were mainly enriched in plant hormone signal transduction, sugar metabolism, amino acid metabolism, energy metabolism and lipid metabolism pathways as well as in LSPs [46]. In this study, the DEGs and DMs of *S. alopecuroides* in different periods of salt stress were analyzed using transcriptomic and metabolomic analyses, and the significantly enriched KEGG pathways were determined; these were mainly KO00500, KO00940, KO00941, and KO04075. These results corroborate those of other studies that employed transcriptomic, proteomic, and metabolomic analyses to study the responses of plants to salt stress. This study indicates that the response process of *S. alopecuroides* to salt stress is similar to that of other plants. Due to its strong ability to adapt to harsh environments, we suspect that even with the same reaction mechanism and metabolic process, certain differences exist between *S. alopecuroides* and other species. We found that the DEGs and DMs obtained in the PHB pathway in *S. alopecuroides* are highly correlated and change significantly under salt stress conditions, which provides us with new research directions.

Phenylpropane metabolism consists mainly of two parts: the lignin and flavonoid metabolism pathways. As the main component of plant cell walls, lignin can increase plant resistance to biotic and abiotic stresses [60]. The miR397 small regulatory RNA *(MicroRNA397)* in Arabidopsis thaliana inhibits the expression of the LAC gene, resulting in decreased lignin content and increased salt sensitivity [61]. The co-transformation of the *PaSOD* and *RaAPX* genes increases the lignin content of *A. thaliana* and enhances its salt tolerance [62]. We found that Sinapyl Alcohol, a precursor of lignin, was continuously upregulated under salt stress conditions in *S. alopecuroides*, and other related metabolites were also continuously upregulated with salt stress conditions. Although the p-coumaroyl and coniferyl alcohol lignin precursors were not detected in our results, we found that genes related to synthetic lignin were also significantly upregulated. This indicated that *S. alopecuroides* could resist salt stress damage by increasing the lignin content. The overexpression of celery NAC (NAM/ATAF/CUC) transcription factor *AgNAC1* in Arabidopsis increases the activity of SOD and POD in drought and salt stress, increases the lignin content, and improves the drought and salt tolerance of *A. thaliana* [60]. The overexpression of NAC transcription factor *BpNAC012* in *Betula platyphylla* increases the expression of the lignin content synthesis gene and enhances its stress resistance [63]. The apple MYB transcription factor *MdMYB46* is overexpressed in *A. thaliana* and Malus pumila and promotes the biosynthesis of secondary cell walls and lignin by directly binding to the promoters of genes related to lignin biosynthesis, thereby improving its salt tolerance [64]. The rice bHLH transcription factor *OsbHLH034* can play a positive regulatory role in the jasmonic acid (JA)-mediated defense response by increasing the lignin content [65]. NAC, MYB (v_myb avian myeloblastosis viral oncogene homolog), and bHLH (basic/helix-loop-helix) transcription factors have been found in other plants to increase the salt tolerance of transgenic plants by increasing the lignin content [62,63,64,65]. In this study, we found that multiple genes *SaPAL, SaC4H, SaCOMT, Sa4CL,* and *SaPOD* involved in lignin synthesis were significantly upregulated under salt stress, and the corresponding metabolites were also upregulated (Figure 6). Therefore, we suspect that genes related to the PHB pathway in *S. alopecuroides* may also be regulated by NAC, MYB, bHLH, and other transcription factors to achieve their functions (Figure 9). The overexpression of the *Fraxinus mandshurica* 4CL gene *Fm4CL2* in tobacco plants can increase their lignin content and significantly improve their tolerance to drought and osmosis [66]. Five 4CL-related genes, *Sa4CL1, Sa4CL2, Sa4CL3, Sa4CL4,* and *Sa4CL5,* were obtained from *S. alopecuroides*, and their expression trends were the same; their expression levels were higher at 4 and 72 h under salt stress (Figure 6). The 4 and 72 h salt treatments were the periods when the RWC of *S. alopecuroides* had obvious changes, which indicated that *Sa4CL* of *S. alopecuroides* might mainly participate in the process of osmotic conditions. The leucaena cinnamic acid coenzyme α-reductase CCR gene is induced by salt stress, which promotes the accumulation of lignin content and improves its stress resistance [67]. The rice cinnamic acid coenzyme α-reductase CCR gene plays a key role in the defense process of rice against Magnaporthe grisea, ultraviolet radiation, and high salinity [68]. The CCR gene is a key enzyme gene for lignin synthesis in the PHB pathway. Our results show that the *SaCCR* from *S. alopecuroides* was significantly upregulated after salt stress (Figure 6 and Figure 8), reaching more than a 200% increase at 24 and 48 h. It was detected that the directly produced product caffeic aldehyde continued to rise under salt stress conditions. This indicates that the *SaCCR* of *S. alopecuroides* actively responds to salt stress. The overexpression of *IbLEA14* in sweet potato plants enhances the tolerance of transgenic calli to drought and salt stress; additionally, the expression of the cinnamic alcohol dehydrogenase gene is activated under salt stress, which may regulate the response to salt stress by enhancing lignification [15]. The *SaCADH* of *S. alopecuroides* was upregulated at various salt stress periods to synthesize lignin precursors (Figure 6 and Figure 8). The accumulation of BR-mediated lignin content in garlic significantly improves its salt tolerance [69]. We suspect that plant hormones may also participate in the regulation of lignin synthesis to resist salt stress. In *S. alopecuroides*, we can further detect the changes of plant hormones after salt stress to explore the role of endogenous hormones in the process of lignin synthesis. In this study, we identified eight gene families in the LSP; these families contributed significantly to the salt stress response process of *S. alopecuroides*. This result indicates that *S. alopecuroides* may increase its resistance to salt stress by increasing its lignin content, which provides a reference for the further exploration and discovery of salt tolerance genes.

As important secondary metabolites of plants, flavonoids play an antioxidant role in the response of plants to abiotic stress [70]. The soybean MYB transcription factor *GmMYB173* can recognize the promoter of the soybean flavonoid synthase gene *GmCHS5*, further enhance the accumulation of dihydroxy B-ring flavonoids, and promote salt tolerance in soybean plants [70]. The overexpression of soybean *GmMYB183* in the roots of soybean plants can promote the accumulation of ononin, which is a B-ring flavonoid with a negative hydroxyl group, and negatively regulates soybean salt tolerance [71]. Soybean heat shock factor *HSFB2b* promotes flavonoid accumulation by activating FLB genes and inhibits the soybean *GmNAC2* gene to release genes in the flavonoid biosynthetic pathway, thereby promoting flavonoid accumulation and improving salt tolerance [72]. Tobacco NtMYB4 negatively regulates the expression of *NtCHS1*, leading to reduced flavonoid accumulation, reduced ROS scavenging ability, and reduced salt tolerance [73]. Antirrhinum bHLH transcription factor gene *AmDEL* improves plants’ tolerance to abiotic stress by increasing FSP *genes*, proline synthesis genes, and ROS scavenging genes in transgenic A. thaliana [74]. Genes related to flavonoid biosynthesis are regulated by MYB, NAC, and bHLH transcription factors, both positive and negative. The obtained *SaHCT, SaCYM, SaFHD,* and *SaCHS* were upregulated under salt stress conditions (Figure 7). In addition, *SaCHIS1* and *SaCHIS2* were upregulated, and *SaCHIS3* was downregulated. The flavonoid synthesis-related genes in *S. alopecuroides* might be regulated by corresponding transcription factors (Figure 7). The overexpression of *AtDFR* in cauliflower can increase the accumulation of anthocyanins and improve the plant’s tolerance to drought and salinity [75]. Rice sulfoquinovosyltransferase SQD2.1 enhances rice plants’ tolerance to osmotic stress by mediating flavonoid glycosylation [76,77]. Reaumuria trigyna leucoanthocyanidin dioxygenase gene *RtLDOX* can replace dioxygenase flavanone 3-hydroxylase (F3H) and convert naringenin into dihydrokaemferol to increase the flavonoid content and plant biomass and further enhance the antioxidant activity of plants [78]. We obtained F3H-related genes from *S. alopecuroides*, named SaFHD, and their expression levels were higher at 24 and 48 h under salt stress (Figure 7). There was no significant change in the antioxidant enzyme content of *S. alopecuroides* under salt stress for 24 and 48 h, indicating that the expression of the FHD gene may be involved in the process of ROS scavenging (Figure 1). The overexpression of *SsMAX2*, a key gene for strigolactone (SL) signaling in the perennial oil plant *Sapium sebiferum*, can enhance the biosynthesis of anthocyanin, promote the expression of antioxidant enzyme genes, and increase the salt tolerance of *A. thaliana* [79]. Exogenous abscisic acid (ABA) and a methyl jasmonate treatment can upregulate the expression of Antarctic moss antioxidant enzyme genes and flavonoid synthesis genes, and improve the plants’ response to high salt stress [71]. In this study, we identified 23 DEGs in the FSP pathway of *S. alopecuroides*, which play an important role in the FSP process. The obtained DEGs were significantly upregulated under salt stress conditions and may be regulated by transcription factors and plant hormones during this process. This needs to be further explored and verified in *S. alopecuroides.*

In summary, the process of *S. alopecuroides* in response to salt stress may be the accumulation of MDA, ROS, soluble sugar, and other substances in a short period of time under salt stress, and signal transduction activates the expression of transcription factors such as MYB, NAC, and bHLH. It further regulates the expression of genes related to lignin and flavonoid biosynthesis; this may activate the salt-response antioxidant enzyme mechanism of *S. alopecuroides*, causing an increase in the content of lignin and flavonoids and improving the tolerance of *S. alopecuroides* to salt stress.

## 4. Material and Methods

### 4.1. Plant Growth and Salt Stress

*Sophora alopecuroides* seeds were collected from the Korla region of Xinjiang, China [32]. Fifteen grams of *S. alopecuroides* seeds were treated with 98% H_2_SO_4_ for 10 min and soaked in distilled water for 16–24 h. Next, they were sown in soil (peat: vermiculite = 1:1), cultivated in a climate room (day/night: 16 h/8 h, 25 °C/22 °C, Changchun City, Jilin Province, China) for germination, and transferred to 1/8 Hoagland’s nutrient solution three weeks later. Then, the plants were treated with 1.2% NaCl (prepared with 1/8 Hoagland’s nutrient solution), harvested at 0, 4, 24, 48, and 72 h, and stored at −80 °C.

### 4.2. Measurement and Analysis of Physiological Indices

The leaf tissues of *S. alopecuroides* that were treated with 1.2% NaCl were used for physiological index determination. For the relative water content measurements, we measured the fresh weight (FW), the saturated weight (SW) (12 h after immersion in distilled water at 4 °C), and the dry weight (DW) (by drying in an oven at 85 °C) [80]. The RWC percentage was calculated by the following formula: FW-DW/(SW-DW). For the determination of various physiological indicators, we used the constant method biochemical kit (Solarbio Life Science, Beijing, China). This kit included chlorophyll (Item No. BC0990), MDA (Item No. BC0020), soluble sugar (Item No. BC0030), proline (Item No. BC0290), and hydrogen peroxide (H_2_O_2_) content determination kits (Item No. BC3590) as well as POD (Item No. BC0090), CAT (Item No. BC0200), and SOD activity detection kits (Item No. BC0170). In the treatment, the Na^+^ and K^+^ contents were determined by digesting the ashed sample and by flame photometry.

### 4.3. Transcriptome Sequencing and Data Analysis

To investigate the changes in the transcriptional level of *S. alopecuroides* at different salt stress time points, total RNA was extracted from the roots of *S. alopecuroides* treated with 1.2% NaCl for 0, 4, 24, 48, and 72 h, with three replicates in each group. RNA integrity was assessed using the RNA Nano 6000 Assay Kit of the Bioanalyzer 2100 system (Agilent Technologies, Santa Clara, CA, USA). Library construction and quality inspection refer to the methods provided by Novogene Co., Ltd. In terms of library construction, the different libraries were pooled according to the effective concentration and the target offline data volume for Illumina sequencing. Data quality control must be performed before analyzing the sequencing results. To ensure the quality and reliability of the data analysis, it is necessary to filter the original data to obtain clean data. The Trinity software was used for transcript assembly. Corset hierarchical clustering was performed for splicing. BUSCO software was used to evaluate the splicing quality of Trinity.fasta, unigene.fa, and cluster.fasta.

The gene function was annotated based on the following databases: Nr (NCBI non-redundant protein sequences); Nt (NCBI non-redundant nucleotide sequences); Pfam (Protein family); KOG/COG (Clusters of Orthologous Groups of proteins); Swiss-Prot (a manually annotated and reviewed protein sequence database); KO (KEGG Ortholog database); GO (Gene Ontology). The differential expression analysis of two conditions/groups was performed using the DESeq2 R package (1.20.0). DESeq2 provides statistical routines for determining differential expression in digital gene expression data using a model based on a negative binomial distribution. The resulting *p*-values were adjusted using the Benjamini–Hochberg approach for controlling the false discovery rate. Genes with an adjusted *p*-value < 0.05, found by DESeq2, were assigned as differentially expressed. Gene Ontology (GO) enrichment analysis of differentially expressed genes was performed using the clusterProfiler R package, in which gene length bias was corrected. We used the clusterProfiler R package to test the statistical enrichment of differentially expressed genes in KEGG pathways. The PPI (protein–protein interactions) analysis of differentially expressed genes was based on the STRING database, which contains known and predicted protein–protein interactions.

### 4.4. Metabolite Profiling Analysis

In the present study, the root tissues of *S. alopecuroides* treated with 1.2% NaCl for 0, 24, 48, and 72 h were used to study the metabolic level of *S. alopecuroides* under salt stress. Each group had six biological replicates for metabolite determination. Tissues (100 mg) were individually ground with liquid nitrogen, and the homogenate was re-suspended with pre-chilled 80% methanol, and 0.1% formic acid by vortexing. The supernatant was injected into an LC-MS/MS system [80]. UHPLC-MS/MS analyses were performed using a Vanquish UHPLC system (ThermoFisher, Dreieich, Germany) coupled with an Orbitrap Q ExactiveTM HF mass spectrometer (ThermoFisher) at Novogene Co., Ltd. (Beijing, China). The raw data files generated by UHPLC-MS/MS were processed using the Compound Discoverer 3.1 (CD3.1; ThermoFisher) to perform peak alignment, peak picking, and quantitation analyses for each metabolite. The normalized data were used to predict the molecular formula based on the additive ions, molecular ion peaks, and fragment ions. Then, the peaks were matched with the mzCloud (https://www.mzcloud.org/, accessed on 23 October 2020), mzVault, and MassList databases to obtain accurate qualitative and quantitative results. Statistical analyses were performed using the statistical software R (R version R-3.4.3), Python (Python 2.7.6 version), and CentOS (CentOS release 6.6); when data were not normally distributed, normal transformations were attempted using the area normalization method.

These metabolites were annotated using the KEGG (https://www.genome.jp/kegg/pathway.html, accessed on 23 October 2020), HMDB (https://hmdb.ca/metabolites, accessed on 23 October 2020), and LIPIDMaps databases (http://www.lipidmaps.org/, accessed on 23 October 2020). Principal component analysis (PCA) and partial least squares discriminant analysis (PLS-DA) were performed using metaX (a flexible and comprehensive piece of software for processing metabolomic data). We employed univariate analysis (*t*-test) to calculate the statistical significance (*p*-value). Metabolites with VIP > 1, a *p*-value < 0.05, and FC ≥ 2 or ≤ 0.5, were considered differential metabolites. Volcano plots were used to filter the metabolites of interest based on the log2 (FC) and log10 (*p*-value) of metabolites. The functions of these metabolites and metabolic pathways were analyzed using the KEGG database. Next, we performed the metabolic pathway enrichment of differential metabolites. When the ratios were satisfied by x/n > y/N, the metabolic pathway was considered as being enriched, while when the *p*-value of the metabolic pathway was <0.05, the enrichment of the metabolic pathway was considered as statistically significant.

### 4.5. Metabolome and Transcriptome Co-Expression Network Analysis

The correlation analysis was performed using Pearson’s correlation coefficient in the R language mixOmics package to calculate the correlation coefficient r2 and the *p*-value of differentially expressed genes and differential metabolites. All differentially expressed genes and differential metabolites obtained in T24_vs_CK, T48_vs_CK, and T72_vs_CK were mapped to the KEGG database to obtain their common pathway information and perform statistical analysis.

### 4.6. The qRT-PCR Validation for DEGs

Based on the results of the joint transcriptomic and metabolomic analysis, we selected 16 differentially expressed genes for qRT-PCR analysis and verification. The specific method and experimental system used have been described in previous reports [32]. Additionally, the data analysis and calculation of the FC values were consistent with those of a previous report [32]. The qRT-PCR primer information is presented in Appendix A.

### 4.7. Statistical Analysis

The index measurement and analysis were performed with at least three biological replicates, and statistical analyses were performed with three or more technical replicates. Statistical analysis and plotting were carried out using Microsoft Excel 2010 and SPSS (SPSS. Inc., USA, version 18.0), and statistical comparisons were performed using *t*-test in SPSS 18.0. The least significant difference (LSD) multi-range test was used to compare the means. Mapping data are presented as mean ± SE, the double asterisk (**) indicates significance at *p* < 0.01 and the single asterisk (*) indicates significant difference at *p* < 0.05.

## Figures and Tables

**Figure 1 ijms-22-02399-f001:**
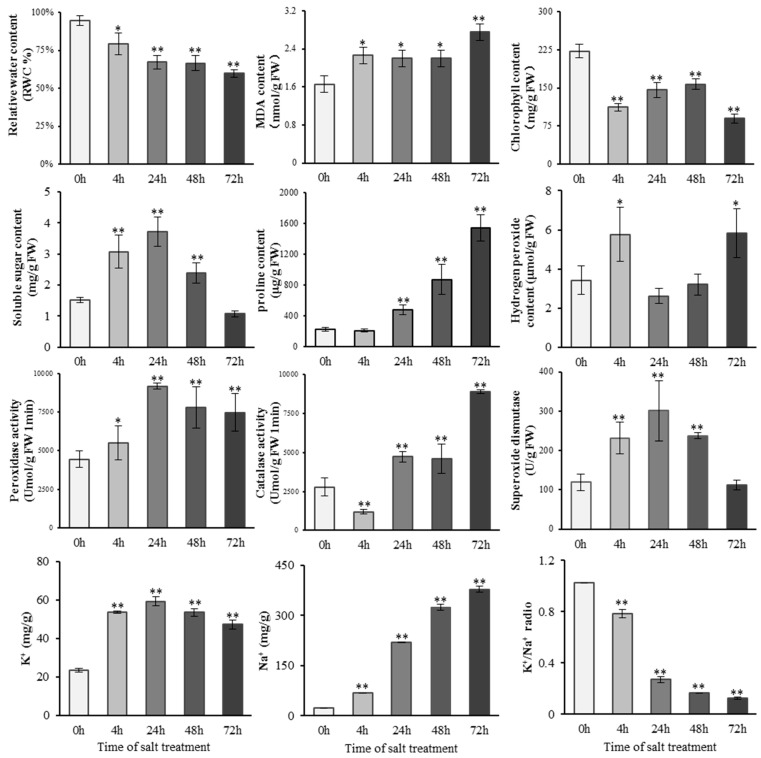
Determination of physiological indexes of *S. alopecuroides* after salt stress. Vertical bar indicates mean + SD calculated from four replicates. Each variable was statistically compared with the control (*t*-test analysis) (** *p* < 0.01, * *p* < 0.05).

**Figure 2 ijms-22-02399-f002:**
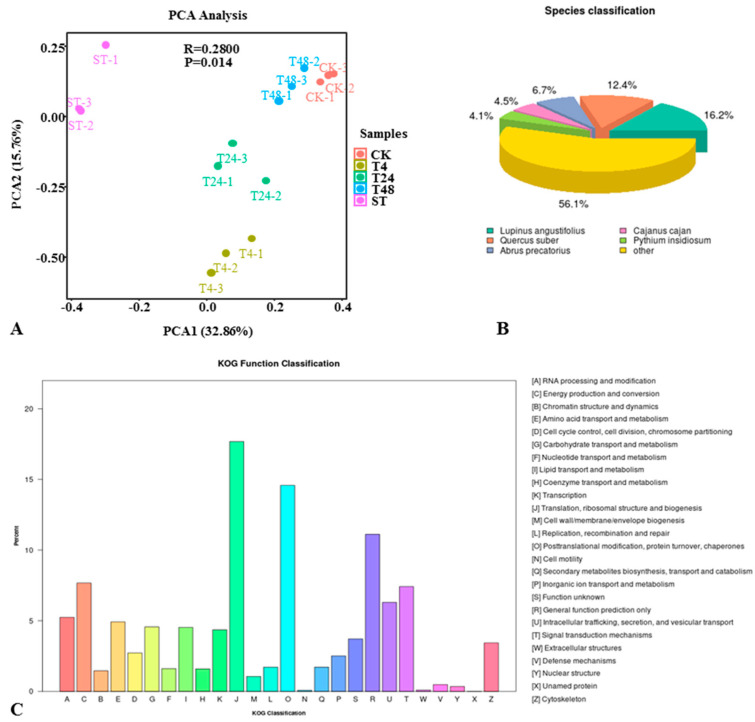
Overview of a time course of *S. alopecuroides* transcriptome responses to salt stress. (**A**) Principal component analysis (PCA) plots of transcripts identified by RNA-seq of salt-stressed *S. alopecuroides* roots at 0, 4, 24, 48, and 72 h after stress. (**B**) Species classification statistics map of the transcriptome NR (NCBI non-redundant protein sequences) library comparison of *S. alopecuroides*. (**C**) KOG (euKaryotic Ortholog Groups) annotated classification chart. The X-axis lists the names of the 26 KOG groups, and the Y-axis has the proportion of the number of genes annotated to the group to the total number of genes annotated.

**Figure 3 ijms-22-02399-f003:**
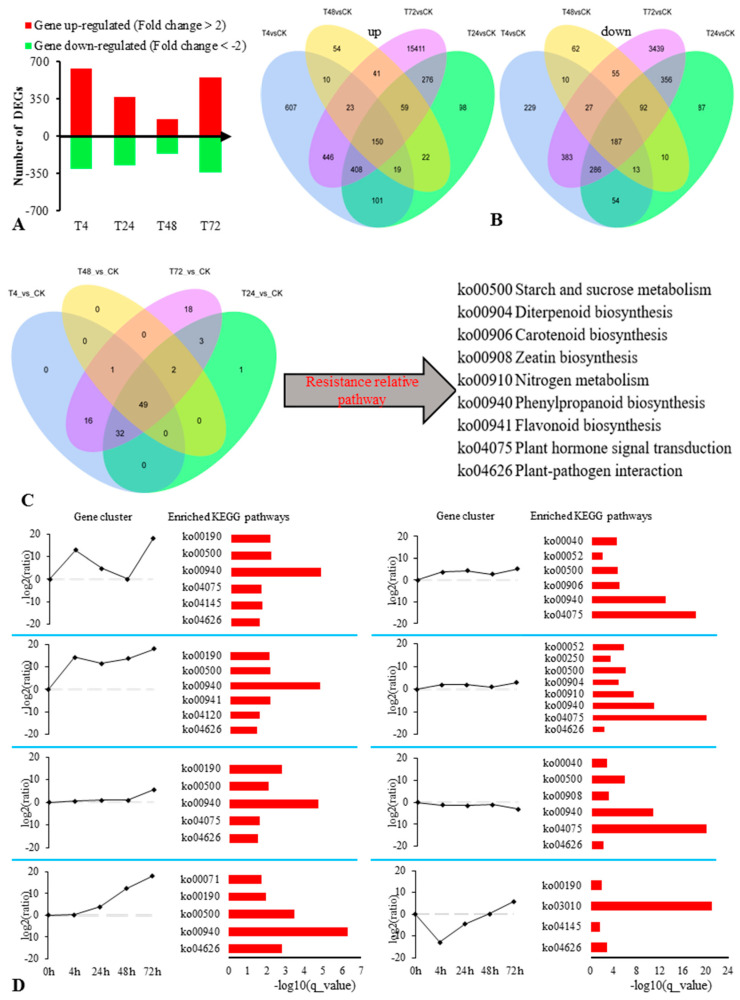
Overview of *S. alopecuroides* differentially expressed genes (DEGs) annotating the KEGG (Kyoto Encyclopedia of Genes and Genomes) pathway response time with salt stress. (**A**) Number of individual transcripts significantly upregulated or downregulated at each time point. (**B**) Venn diagram illustrating the number of transcripts upregulated or downregulated by salt stress over the time course. (**C**) Venn diagram showing the overlap of common and unique pathways present in the transcriptome following 4, 24, 48, and 72 h post salt stress. (**D**) Time sequence transcriptomic analysis of salt stress inducing significant DEGs in the roots of *S. alopecuroides.* The KEGG pathways for each profile are listed on the right. Enrichment scores are shown as −log10(q).

**Figure 4 ijms-22-02399-f004:**
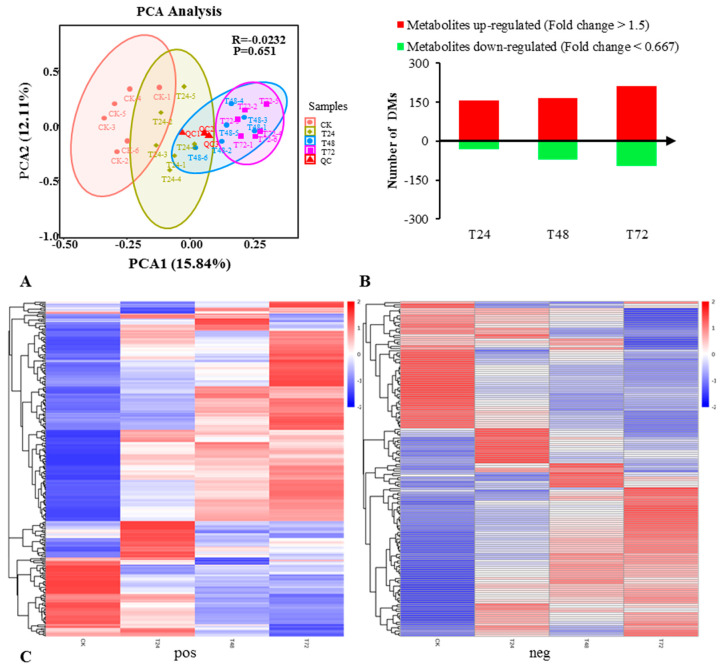
Overview of a time course of *S. alopecuroides* metabolome responses to salt stress. (**A**) PCA plots of metabolites identified by UHPLC (Ultra Performance Liquid Chromatography and High Resolution Mass Spectrometry)-MS/MS of salt-stressed *S. alopecuroides* roots at 0, 24, 48, and 72 h after stress. (**B**) Number of individual metabolites significantly upregulated or downregulated at each time point. (**C**) Clustering heat map of total difference metabolites (the left frame is the positive ion mode, the right frame is the negative ion mode). The longitudinal direction is the clustering of samples, and the shorter the cluster branches, the higher the similarity.

**Figure 5 ijms-22-02399-f005:**
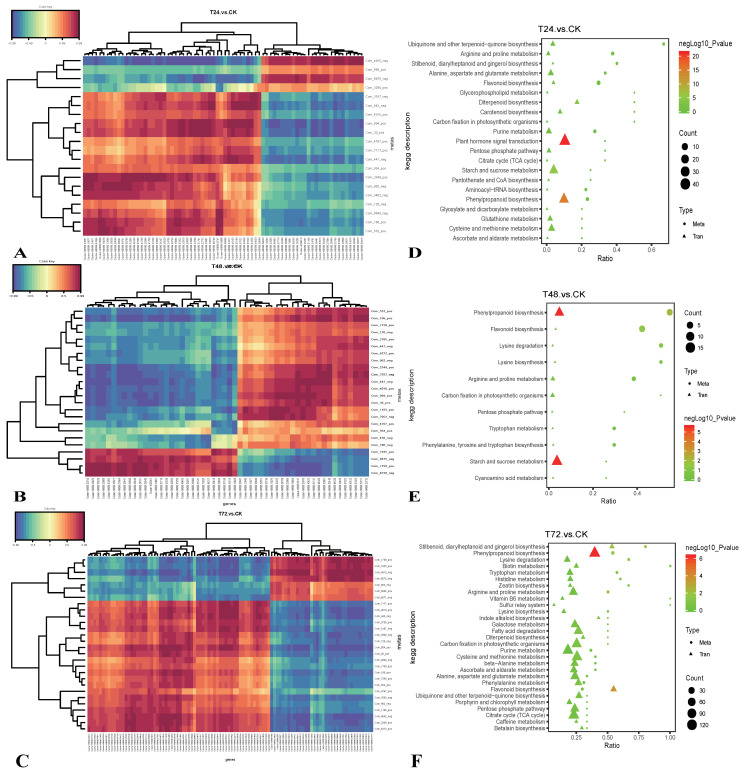
Overview of the correlation between differential metabolites (DMs) and DEGs. (**A**–**C**) Heat map of correlation analysis between DMs and DEGs (pos + neg). The vertical direction represents the DEGs cluster, and the horizontal direction represents the DMs cluster. The shorter the cluster branches, the higher the similarity. Blue indicates negative correlation, and red indicates positive correlation. (**D**–**F**) Correlation analysis of DMs and DEGs KEGG pathway (pos + neg). The abscissa is the ratio of the DMs and DEGs enriched in the pathway to the number of metabolites or genes annotated in the pathway (Ratio), and the ordinate is the KEGG pathway that is jointly enriched by the metabolome–transcriptome. Count: The number of metabolites or genes enriched in the pathway.

**Figure 6 ijms-22-02399-f006:**
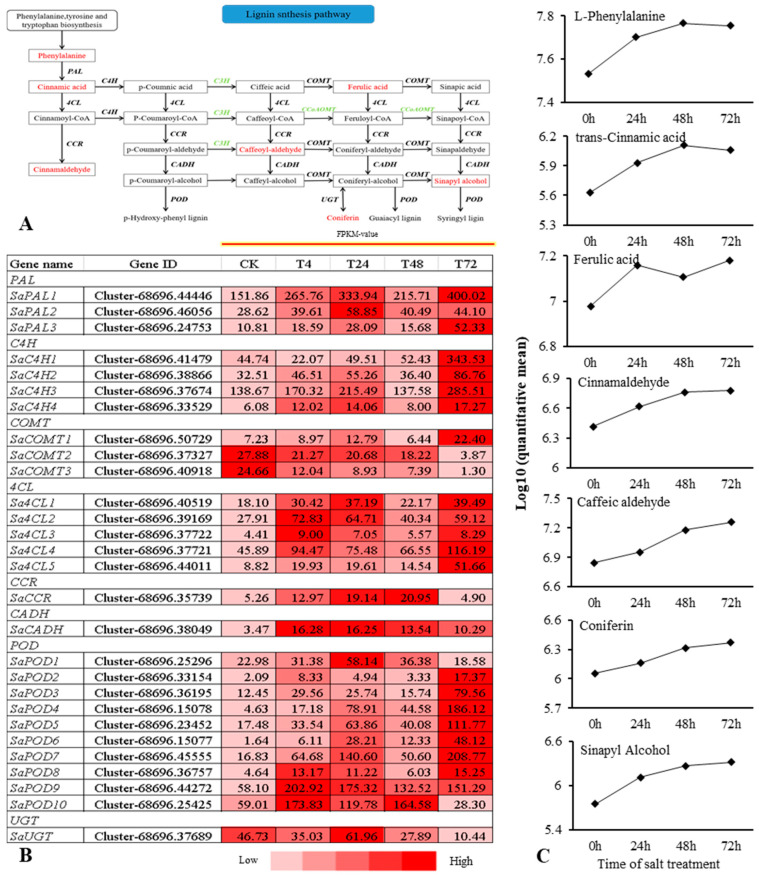
Overview of the relationship between DEGs and DMs in the lignin synthesis pathway (LSP) of *S. alopecuroides* under salt stress. (**A**) Overview of lignin biosynthesis. (**B**) Heat map of lignin biosynthesis-related gene expression. Values are Average FPKM value of each sample in each group. PAL, phenylalanine ammonia-lyase; C4H, trans-cinnamate 4-monooxygenase; COMT, caffeic acid 3-O-methyltransferase; 4CL, 4-coumarate–CoA ligase; CCR, cinnamoyl-CoA reductase; CADH, cinnamyl alcohol dehydrogenase; POD, peroxidase; UGT, coniferyl-alcohol glucosyltransferase. (**C**) The trend of lignin synthesis pathway DMs changes with salt stress. Expression scores are shown as log10(q).

**Figure 7 ijms-22-02399-f007:**
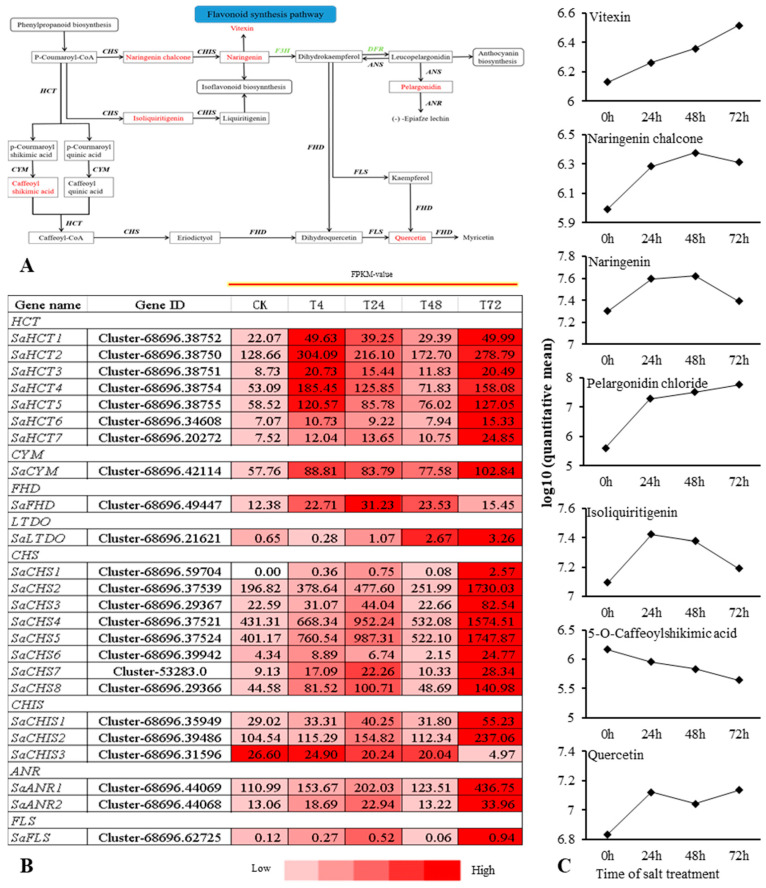
Overview of the relationship between DEGs and DMs in the FSP of *S. alopecuroides* under salt stress. (**A**) Overview of flavonoid biosynthesis. (**B**) Heat map of flavonoid biosynthesis-related gene expression. Values are the average FPKM value of each sample in each group. HCT, shikimate O-hydroxycinnamoyltransferase; CHS, chalcone synthase; CHIS, chalcone isomerase; CYM, coumaroylquinate (coumaroylshikimate) 3′-monooxygenase; FHD, flavonoid 3′,5′-hydroxylase; FLS, flavonol synthase; ANR, anthocyanidin reductase; ANS, leucoanthocyanidin dioxygenase. (**C**) The trend of flavonoid synthesis pathway DMs changes with salt stress. Expression scores are shown as log10(q).

**Figure 8 ijms-22-02399-f008:**
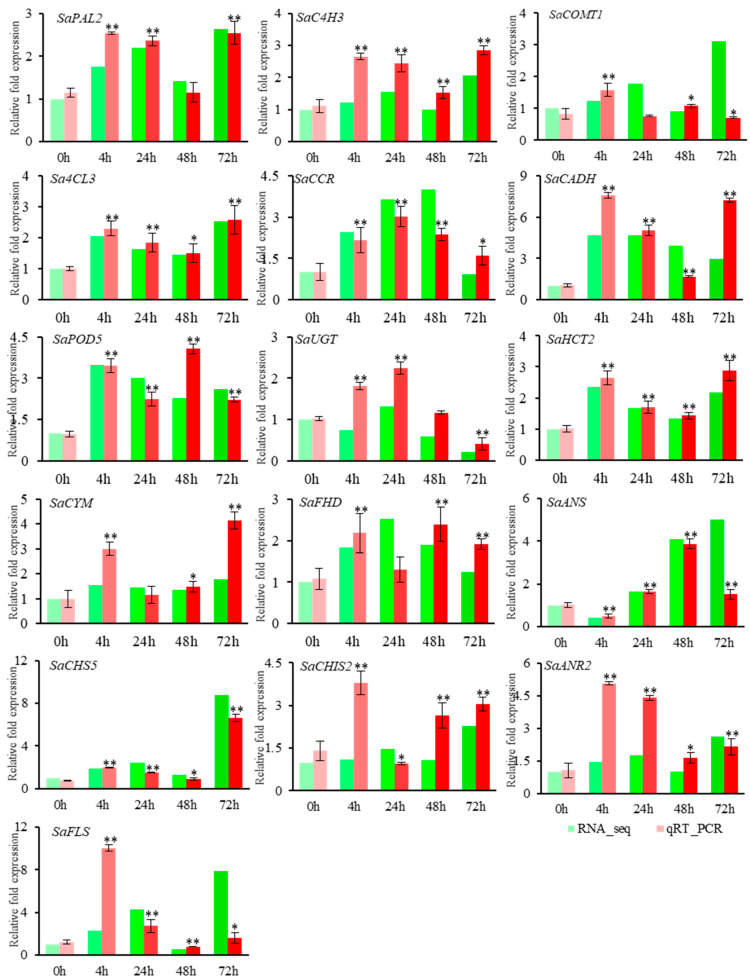
QRT-PCR verification of DEGs. The relative gene expression levels under 1.2% NaCl treatment at different periods. Vertical bar indicates mean + SD calculated from three replicates. Each variable was statistically compared with the control (*t*-test analysis) (** *p* < 0.01 * *p* < 0.05).

**Figure 9 ijms-22-02399-f009:**
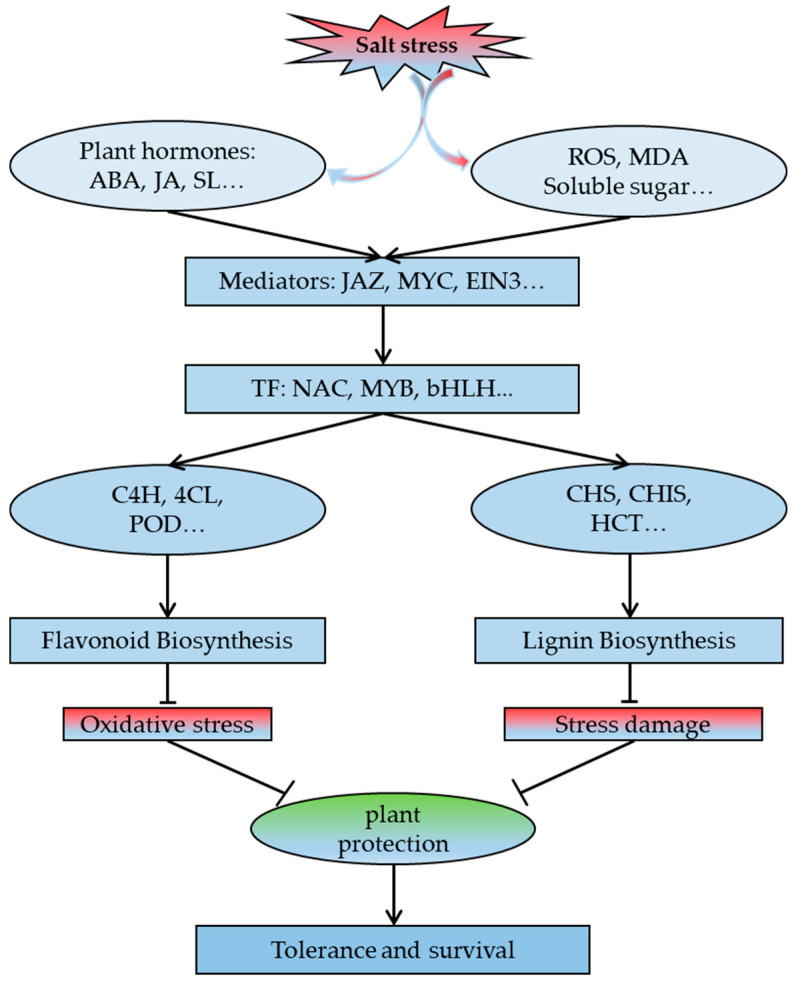
Signal transduction and regulation of secondary metabolism in response to salt stress in *S. alopecuroides.* The arrow indicates that the previous item activates the latter item, and the non-arrow indicates that the previous item inhibits the latter item.

**Table 1 ijms-22-02399-t001:** Statistics of success rate of gene annotation.

Database	Number of Unigenes	Percentage (%)
Annotated in NR	80,384	55.8
Annotated in NT	72,447	50.29
Annotated in KO	31,769	22.05
Annotated in SwissProt	59,963	41.62
Annotated in PFAM	61,906	42.97
Annotated in GO	61,901	42.97
Annotated in KOG	27,796	19.29
Annotated in all Databases	12,155	8.43
Annotated in at least one Database	105,318	73.11
Total Unigenes	144,051	100

## Data Availability

Not applicable.

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
