# Peer review of "Combined Transcriptomic and Metabolomic Analysis Reveals the Role of Phenylpropanoid Biosynthesis Pathway in the Salt Tolerance Process of *Sophora alopecuroides"

_ijms, 2021, doi:10.3390/ijms22052399_

Round 1

Reviewer 1 Report

Authors reported "Combined transcriptomic and metabolomic analysis reveals the role of phenylpropanoid biosynthesis pathway in the salt tolerance process of Sophora alopecuroides". Authors not only investigated to clarify the physiological status after salt treatment to natural variety from Sophora alopecuroides, but also authors studied the identification and quantification of transcripts from de novo assembly of RNA-seq under salt stress. This report provides many information and it likely gives the knowledges for researchers who are working in Sophora alopecuroides and involving in an environmental stress but this report has shortcoming that need to be addressed before this can be published in a journal. In particular, I have two concerns. What is “Sophora alopecuroides”? Additional information involved in Sophora alopecuroides should be provided. For example, if authors could give any photos of this plant such as photos after salt stress and before salt stress, it is good for me and others. As second, samples origin was different among experiments of physiological status, transcriptional experiments and discussion. Is it possible to prepare separately the leaves and root in discussion?  I hope authors provides the information such as belows…

Introduction

Lane101 Please provide genome information (i.e. genome size) of Sophora alopecuroides.

Lane101 Please give any photo about Sophora alopecuroides.

Lane 104 Is there any reports of Sophora alopecuroides which seem to be resistant to salt damage and acidic soil. I was not sure how resistant this plant is to salt damage compared to other plants. Is there any literature that compares, and it may be very desirable.

Figure 1 Space between catalase and activity in y axis

Figure 1 For Statistical comparisons, authors used one-way ANOVA for statistical analysis. Is this correct? What data were you comparing to find the p-value? Isn't it usually shown as the a, b and ab on bar?

Figure 3 Resolution should be improved. Words is samll.

Lane 538 Is there any cultivation name?

Lane 541 What is cell meaning? Here may be plants?

Lane 545 italic format “S. alopecuroides”

Reviewer 2 Report

In this work Youcheng Zhu et al. investigated the role of phenylpropanoid biosynthesis pathway in the salt tolerance process of Sophora Alopecuroides through a combined transcriptomic and metabolomic analysis. The work is original. Despite its complexity it is clear, well organized, well written supported by a strong statistical analysis. One minor points is listed below.

1) lines 546-547: “For the relative water content measurements we measured the fresh weight (FW), the saturated weight (SW) (12 h after immersion in distilled water at 4 °C)”. This procedure was employed for rinsing the leaf samples? Why  the authors employed this procedure (12 f immersion) instead of a, for example,  3 time “fast” rinsing  etc… why 85°C? Is there the risk of removing also “bound” water o denaturing proteins? Please give references.

Author Response

Thank you for critical review the manuscript.

For the determination of the relative water content of plant leaf tissue, we refer to previous studies. The reference was cited in the material (line 545).

  1. Immersion in distilled water was employed for measuring the saturated weight (SW).
  2. Normally, the dry weight of plant tissue was obtained by drying the sample at 80-85℃. The control and treated samples were under same conditon in the experiment. The effect of protein denaturation can be normalized by control.

[81] Meher.; Shivakrishna, P.; Ashok, R.K.; Manohar, R.D. Effect of PEG-6000 imposed drought stress on RNA content, relative water content (RWC), and chlorophyll content in peanut leaves and roots. Saudi J Biol Sci. 2018, 25, 285-289.